# Patterns of testing in the extensive Danish national SARS-CoV-2 test set-up

**Mie Agermose Gram**[1]*, **Nina Steenhard**[2], **Arieh Sierra Cohen**[2], **Anne-Marie Vangsted**[2], **Kåre Mølbak**[3,4], **Thøger Gorm Jensen**[5,6], **Christian Holm Hansen**[1], **Steen Ethelberg**[1,7]

**1** Department of Infectious Disease Epidemiology and Prevention, Statens Serum Institut, Copenhagen S, Denmark, **2** TestCentre Denmark, Statens Serum Institut, Copenhagen S, Denmark, **3** Division of Infectious Disease Preparedness, Statens Serum Institut, Copenhagen S, Denmark, **4** Department of Veterinary and Animal Sciences, Faculty of Health and Medical Sciences, University of Copenhagen, Frederiksberg C, Denmark, **5** Department of Clinical Microbiology, Odense University Hospital Odense C, Odense, Denmark, **6** Research Unit of Clinical Microbiology, University of Southern Denmark, Odense, Denmark, **7** Department of Public Health, Global Health Section, University of Copenhagen, Copenhagen K, Denmark

* miag@ssi.dk

## Abstract

### Background

The Danish national SARS-CoV-2 mass test system was among the most ambitious worldwide. We describe its set-up and analyse differences in patterns of testing per demography and time period in relation to the three waves of SARS-CoV-2 transmission in Denmark.

### Methods

We included all reported PCR- and rapid antigen-tests performed between 27 February 2020 and 10 March 2022 among all residents aged 2 years or above. Descriptive statistics and Poisson regression models were used to analyse characteristics of individuals tested for SARS-CoV-2 using a national cohort study design.

### Results

A total of 63.7 million PCR-tests and 60.0 million rapid antigen-tests were performed in the study period, testing 90.9% and 78.8% of the Danish population at least once by PCR or antigen, respectively. Female sex, younger age, Danish heritage and living in the capital area were all factors positively associated with the frequency of PCR-testing. The association between COVID-19 vaccination and PCR-testing changed from negative to positive over time.

### Conclusion

We provide details of the widely available, free-of-charge, national SARS-CoV-2 test system, which served to identify infected individuals, assist isolation of infectious individuals and contact tracing, and thereby mitigating the spread of SARS-CoV-2 in the Danish population. The test system was utilized by nearly the entire population at least once, and widely accepted across different demographic groups. However, demographic differences in the

**Data Availability Statement:** The data material used involve information on every person living in Denmark within the study period. Data cannot be shared publicly because they may only be

accessed and register-coupled within a secure data analysis environment. However, de-identified participant-level data are available for Institutional Data access to members of the scientific and medical community for non-commercial use only. Applications should be submitted to Forskerservice (https://sundhedsdatastyrelsen.dk/da/forskerservice) at The Danish Health Data Authority, where they will be reviewed on the basis of relevance and scientific merit. Data are available now, with no defined end date.

**Funding:** The authors received no specific funding for this work.

**Competing interests:** The authors have declared that no competing interests exist.

test uptake did exist and should be considered in order not to cause biases in studies related to SARS-CoV-2, e.g., studies of transmission and vaccine effectiveness.

## Background

Diagnostic testing to identify individuals infected with the highly transmissible severe acute respiratory syndrome coronavirus 2 (SARS-CoV-2) is important to reduce the spread of the virus [1, 2]. Public health experts have emphasized mass testing of as many individuals as possible—tracking infected individuals, and contact tracing—as an effective strategy to reduce the spread of the virus [1]. During 2020–2022, Denmark set up one of the highest PCR mass testing capacities per capita in the world [3]. The extensive, openly available, free-of-charge, national SARS-CoV-2 test system in Denmark, which was initiated at the beginning of the epidemic (April 2020), was considered by the government to be a prerequisite for an early reopening after the proactive lockdown of 11 March 2020. The system was continuously expanded throughout the pandemic [4]. Rapid antigen-tests were introduced in April 2021. PCR-tests and rapid antigen-tests were available to all whether symptomatic or not and without needing a referral. The main goal in Denmark was to reduce the spread of SARS-CoV-2, protect vulnerable individuals such as nursing home residents and hospitalized patients, avoid overloading the healthcare system and keep society open to the largest possible extent. The test strategy was an important tool to identify infected individuals including non-symptomatic individuals, thereby reducing the spread of SARS-CoV-2 by initiating isolation, and contact tracing.

The national SARS-CoV-2 test system also turned out to be a critical source of data for surveillance and epidemiological analysis of the epidemic, including vaccine effectiveness analyses. For these reasons, and to inform future decisions on test strategies, it is important to describe the system and understand how test results reflect the underlying population. In this paper we provide a description of the set-up of the Danish national SARS-CoV-2 test system and analyse differences in PCR-testing patterns per demographic group and time period in relation to the three main waves of SARS-CoV-2 transmission which occurred in Denmark.

## Methods

### Description of the development of the Danish national SARS-CoV-2 test system

Denmark, a Scandinavian country of 5.86 million inhabitants, choose a two-pronged approach to testing for SARS-CoV-2, where testing was organized in two tracks, the so-called 'healthcare' and 'community' tracks [5]. The healthcare track analysed all samples in the first months of the epidemic and then took a clinical approach with both in- and out-patients tested based on medical referrals, screening before hospitalization, outpatient visits at hospitals or at general practitioners as well as regular screening of healthcare staff. The samples from the healthcare track were analysed by the 10 clinical microbiology laboratories that exist in Denmark, within the regional hospitals, usually around the clock for hospital patients. During 2020, capacity was greatly increased by new instruments and expansion of the staff, the laboratories were mandated to be able to analyse up to 30,000 samples a day.

The second track, the community track, was established in addition to the healthcare track to provide readily available, free-of-charge testing in the local community, on-demand and without referrals. The community track, became responsible for more than 80% of the tests

and its set-up is briefly summarised below. More information, including description of the laboratory procedures and detection of viral variants is found in the S1 File.

Laboratory activities in the community track were centralized and conducted at Statens Serum Institut (SSI), the Danish national institute for infectious disease control which operates under the authority of the Ministry of Health. In April 2020, a high capacity, highly automated laboratory (TestCenter Denmark) was established at SSI in order to handle the large influx of samples. During the pandemic, the laboratory capacity reached 200.000 daily PCR-tests; the mandated performance requirement was that test results for 80% were reported within 24 hours. Oropharyngeal swabs for PCR-testing were taken by trained professionals in a network of testing stations that were established during the pandemic. When operating at full capacity, people had a maximum of 20 km distance from their home to the nearest testing station as per government requirements.

## Data logistics

The testing stations of the community track used a centralized on-line booking system allowing users to book an appointment at any testing station via a single webpage. When booking on-line, users were identified via the existing "NemID" system, which is an app-based all-purpose, compulsory, national electronic identification system consisting of a user ID, a password and a (electronic) code card. Drop-in testing was sometimes available, subject to the demand for testing. All samples were registered electronically using bar code scanning in conjunction with sampling and without paper forms. All samples were registered via WebReq, which is a national online system for medical sample registration [5]. Users were identified by scanning their healthcare ID card. Samples were registered by scanning the 2D barcode on the bottom of the matrix tube (a tube originally used for biobanking and not routine microbiology testing; in the healthcare track tubes were labelled with a unique national sample number). Thus, all sample registration was performed electronically with no manual data entry. All order data was transferred to a database at either SSI for the community track or the hospital laboratory for the healthcare track and was used to match laboratory results with patient data.

All test results were registered on the individual, using the Danish unique personal identification number, which is used in all national registries, enabling individual-level linkages between registries [6]. Results were, depending on the requestor, reported electronically to either the patient's GP or the hospitals electronic patient record and at the same time sent to and registered in the Danish Microbiology Database (MiBa) enabling real-time epidemiological surveillance throughout the epidemic [5]. The SARS-CoV-2 results in MiBa were captured by an online service that allowed individuals to see their own test results via various existing public web-based health interfaces (e.g. www.sundhed.dk) and mobile apps. Upon completed test, the user would receive an SMS or push-up notice, advising the user to log in and see the result. Users that tested positive for variants of concern could then be contacted directly via telephone by the contact tracing group of the Danish Patient Safety Authority.

## Rapid antigen-testing

From February 2021 rapid antigen-test also became part of the Danish community test system. It was funded by the Government, relied on test stations, and the public national data infrastructure. Thus, for public rapid antigen-testing, all test data and results were electronically traceable and thus available to the users, the medical system and for surveillance—national incidence figures were in general based on PCR-tests alone, but antigen-testing used for the contact tracing. A variety of assays were used for rapid antigen-testing, which was performed by several private companies on government contracts offering their services at testing stations

throughout the country. Individuals who received a positive antigen-test result were advised to have it confirmed with a PCR-test in the community track, which also allowed positive samples to undergo WGS.

## Data for the analyses of this paper

The data analysis study presented in this paper was designed as a nationwide register-based cohort study, i.e., with the entire Danish population as cohort members. PCR-tests from both the community track and the healthcare track were included. The proportion of total PCR-test performed was 83.4% and 16.6% in the community and healthcare track, respectively.

Information on date of birth, sex, heritage, vital status, emigration and current residence municipality were obtained from the Danish Civil Registration System (CRS) [6]. Information on vital status and emigration status were used for censoring in the Poisson regression models. Information on all positive, negative and inconclusive PCR-tests and rapid antigen-tests performed by the healthcare track and the society track were obtained from MiBa [5]. Information on administrated COVID-19 vaccines was obtained from the Danish Vaccination Registry [7].

The study population included all residents in Denmark aged 2 years or above (all residents eligible for SARS-CoV-2 tests). The study period was February 27, 2020 to March 10, 2022, i.e. from the first diagnosed case in Denmark until the official end of the period during which testing without medical indication was broadly available. The study consists of two descriptive analyses. The first descriptive analysis provides characteristics of individuals tested 'never', 'rarely', 'commonly' and 'often', stratified by PCR and rapid antigen-tests. The second descriptive analysis provides characteristics of PCR-tested individuals in three sub-periods to account for variation in test capacity and the emergence of new SARS-CoV-2 variants in each period: The periods were defined according to implementation and lifting of restrictions. Period I was from the first positive PCR-test in Denmark until the start of the second lockdown (February 27, 2020 to December 16, 2020). Period II was from the start of the second lockdown until all restrictions were lifted (December 17, 2020 to September 30, 2021); this period corresponds also broadly speaking to the time when vaccines were rolled out. Period III spanned from the reintroduction of restrictions until the implementation of guidelines [8] advising that PCR-testing be done only if symptomatic or at increased risk of a serious course of COVID-19 (November 11, 2021 to March 10, 2022), effectively ending the mass-testing period. We used Poisson regression to provide an incidence rate ratio (IRR) including a 95% confidence interval (CI), for being PCR-tested by sex, age groups, vaccination status, infection status, heritage and type of area. For this analysis, antigen-tests were excluded since during the epidemic, the PCR-test results were the primary basis for surveillance and since PCR-tests have been primarily utilized in epidemiological studies.

## Statistical analysis

Characteristics of individuals that were tested 'never', 'rarely', 'commonly' or 'often' were described using proportions. For both PCR-tests and rapid antigen-tests, individuals were categorized as 'never tested' if they had 0 tests, 'rarely' if they had 1–3 tests, 'commonly' if they had 4–15 tests and 'often' if they had more than 15 tests performed during the entire study period. Rarely, commonly and often were defined by quartiles of number of PCR-test by individuals with one or more PCR-tests. Characteristics of individuals PCR-tested in Period I, II and III were described using number of tested individuals and proportion tested individuals. The included characteristics were sex (female/male), age group at start of study period (10 levels: 2–9 years, 10–19 years, 20–29 years, 30–39 years, 40–49 years, 50–59 years, 60–69 years,

70–79 years, 80–89 years and 90+ years), heritage (four levels: Danish, non-western, western and unknown) and type of area (six levels: capital municipalities, commuter municipalities, metropolitan municipalities, provincial municipalities, rural municipalities and unknown). The definitions of Heritage and type of area were according to Statistics Denmark [9]. The heritage definition in presented in the S1 Table in S1 File.

Crude and adjusted IRR were calculated using Poisson regression models for each of the three periods. The adjusted Poisson regression models were adjusted for the variables above defined, i.e.: age group, sex, vaccination status, SARS-CoV-2 infection status (no infection/previous infected), heritage, and type of area. Individuals were defined as previous infected if they were registered with a positive PCR-test in MiBa otherwise they were defined as not infected. Vaccination status and infection status were included as time-varying covariates in the regression model by splitting time using Lexis expansion. However, for Period I, vaccination status was not analysed as the vaccination program had not yet been initiated. Only the covariate heritage had missing values. The missing values were categorized as unknown and included in analyses.

### Ethics

This article only contains aggregated results and no personal data. For this reason, consent was not needed and not obtained. Further, according to Danish law, ethical approval should not be obtained for anonymized aggregated register-based studies not using biological material (cf. implementing decree, act no. 1338 of 2020-09-01 on research ethics review of health research projects). For the same reasons, the article is not covered by the European General Data Protection Regulation. This work was carried out under the Surveillance auspices of the SSI as regulated in paragraph 222 of the Danish Communicable Disease Act and was therefore not listed with the Danish Data Protection Agency.

### Results

Between February 27, 2020 and March 10, 2022, there were in total 6,086,203 million residents in the Danish population of which 5,844,172 were 2 years or above at the start of this period. During this period, a total of 63.7 million PCR-tests and 60.0 million rapid antigen-tests were registered. The median number of PCR-tests and rapid antigen-tests done per individual was 7. Of the 5,844,172 individuals, 9.1% were never PCR-tested while 21.2% were never rapid antigen-tested. Moreover, 5.4% were never PCR- or rapid antigen-tested. Among those often PCR-tested, the proportion of males was lower compared to females. The proportion of often PCR-tested was 27.8% and 18.9% among females and men, respectively. The proportion of never PCR-tested was 7.3% and 17.5% among individuals vaccinated and not vaccinated during the study period, respectively (Table 1). Furthermore, the proportion of never PCR or antigen-tested were higher among the oldest age groups compared to the youngest age groups (Table 1).

The number of daily samples increased throughout the pandemic from a few thousand to a maximum of over 200,000 in the community track and 30,000 in the healthcare track. The number of PCR-tests peaked in December 2021 and January 2022, while the number of rapid antigen-tests peaked in May and June 2021; the highest number of daily PCR and rapid antigen-tests were 270,248 and 544,361, respectively (Fig 1). The number of PCR-test stations in use also varied over the study period, with a median of 268 available PCR-test stations. During 2020, the number of test stations increased from 18 to 250. The highest number were available between March and June 2021, and between December 2021 and February 2022 (Fig 2).

**Table 1. Characteristics of 5,844,172 individuals tested by PCR-test or rapid antigen-test, by group of test frequency, categorised as: Never, rarely, commonly and often, Denmark, 2020–2022.**

| | PCR-tested | | | | Rapid antigen-tested | | | |
|---|---|---|---|---|---|---|---|---|
| | Never (%) 0 test | Rarely (%) 1–3 test | Commonly (%) 4–15 test | Often (%) >15 test | Never (%) 0 test | Rarely (%) 1–3 test | Commonly (%) 4–15 test | Often (%) >15 test |
| **Total** | 9.1 | 20.2 | 47.3 | 23.4 | 21.2 | 18.6 | 34.2 | 25.9 |
| **Sex** | | | | | | | | |
| Female | 8.2 | 18.1 | 45.9 | 27.8 | 21.2 | 18.3 | 33.4 | 27.1 |
| Male | 10.0 | 22.3 | 48.8 | 18.9 | 21.2 | 19.0 | 35.0 | 24.7 |
| **Age groups (years)** | | | | | | | | |
| 2–9 | 4.6 | 18.8 | 62.2 | 14.4 | 40.2 | 24.4 | 30.2 | 5.2 |
| 10–19 | 2.2 | 11.9 | 61.2 | 24.7 | 4.4 | 6.9 | 27.9 | 60.8 |
| 20–29 | 7.8 | 16.1 | 51.2 | 24.9 | 9.0 | 8.7 | 34.0 | 48.4 |
| 30–39 | 6.9 | 15.5 | 49.1 | 28.5 | 10.8 | 12.9 | 41.0 | 35.4 |
| 40–49 | 5.5 | 14.0 | 45.4 | 35.0 | 11.3 | 15.9 | 42.5 | 30.2 |
| 50–59 | 6.2 | 16.2 | 44.1 | 33.5 | 13.7 | 18.6 | 42.3 | 25.4 |
| 60–69 | 8.6 | 20.1 | 42.0 | 29.4 | 21.0 | 24.0 | 39.2 | 6.4 |
| 70–79 | 14.5 | 29.3 | 43.3 | 12.9 | 30.4 | 29.3 | 33.9 | 6.4 |
| 80–89 | 20.1 | 37.0 | 37.7 | 5.2 | 41.6 | 33.2 | 23.2 | 2.0 |
| ≥90 | 26.5 | 38.5 | 29.7 | 5.3 | 74.1 | 20.1 | 5.6 | 0.2 |
| **Vaccinated with ≥2 doses before March 10 2022** | | | | | | | | |
| Yes | 7.3 | 19.3 | 47.4 | 26.0 | 17.6 | 19.0 | 35.8 | 27.5 |
| No | 17.5 | 24.3 | 47.1 | 11.1 | 38.1 | 16.9 | 26.6 | 18.4 |
| **Heritage** | | | | | | | | |
| Danish | 7.8 | 19.6 | 47.5 | 25.2 | 20.6 | 18.5 | 34.3 | 26.7 |
| Non-west | 9.4 | 24.0 | 51.9 | 14.6 | 19.2 | 20.7 | 36.6 | 23.4 |
| West | 17.3 | 23.4 | 42.5 | 16.8 | 25.3 | 19.1 | 33.1 | 25.5 |
| Unknown | 96.3 | 1.2 | 2.1 | 0.4 | 97.8 | 0.7 | 1.0 | 0.5 |
| **Type of area (municipality)** | | | | | | | | |
| Capital Municipalities | 8.2 | 16.8 | 49.2 | 25.8 | 19.8 | 16.4 | 34.4 | 29.4 |
| Commuter Municipalities | 9.9 | 22.6 | 46.7 | 20.8 | 21.6 | 20.1 | 35.2 | 23.1 |
| Metropolitan Municipalities | 8.6 | 19.0 | 47.8 | 24.6 | 19.2 | 17.3 | 33.6 | 29.9 |
| Provincial Municipalities | 9.0 | 20.9 | 46.9 | 23.2 | 22.0 | 19.7 | 34.3 | 24.0 |
| Rural Municipalities | 10.1 | 22.9 | 45.4 | 21.6 | 23.3 | 20.3 | 33.5 | 23.0 |

The proportion of individuals PCR-tested at least once was 63.7% in Period I, but higher in Period II (76.4%) and III (78.9%) although the opportunity to be rapid antigen-tested also existed in these periods. In all three periods, females, younger age groups and individuals from municipalities in the capital area were more likely to get PCR-tested (Table 2). Especially the older age groups were less likely to get PCR-tested in Period II and III compared to Period I. Among individuals aged 70–79, 53.4% were PCR-tested in Period I compared to 67.7% and 68.6% which were PCR-tested in Period II and III, respectively (Table 2).

The estimates from the adjusted Poisson regression models showed that throughout study period I, II and III, males were less likely to be PCR-tested, (IRR: 0.73–0.78) compared to females (Fig 3 and S2 Table in S1 File). Also, over both Periods I, II and III, individuals with non-Danish heritage had relatively lower test rates. In period I (i.e., most of 2020), individuals aged 2–9 years and the oldest age groups were least likely to be PCR-tested compared to

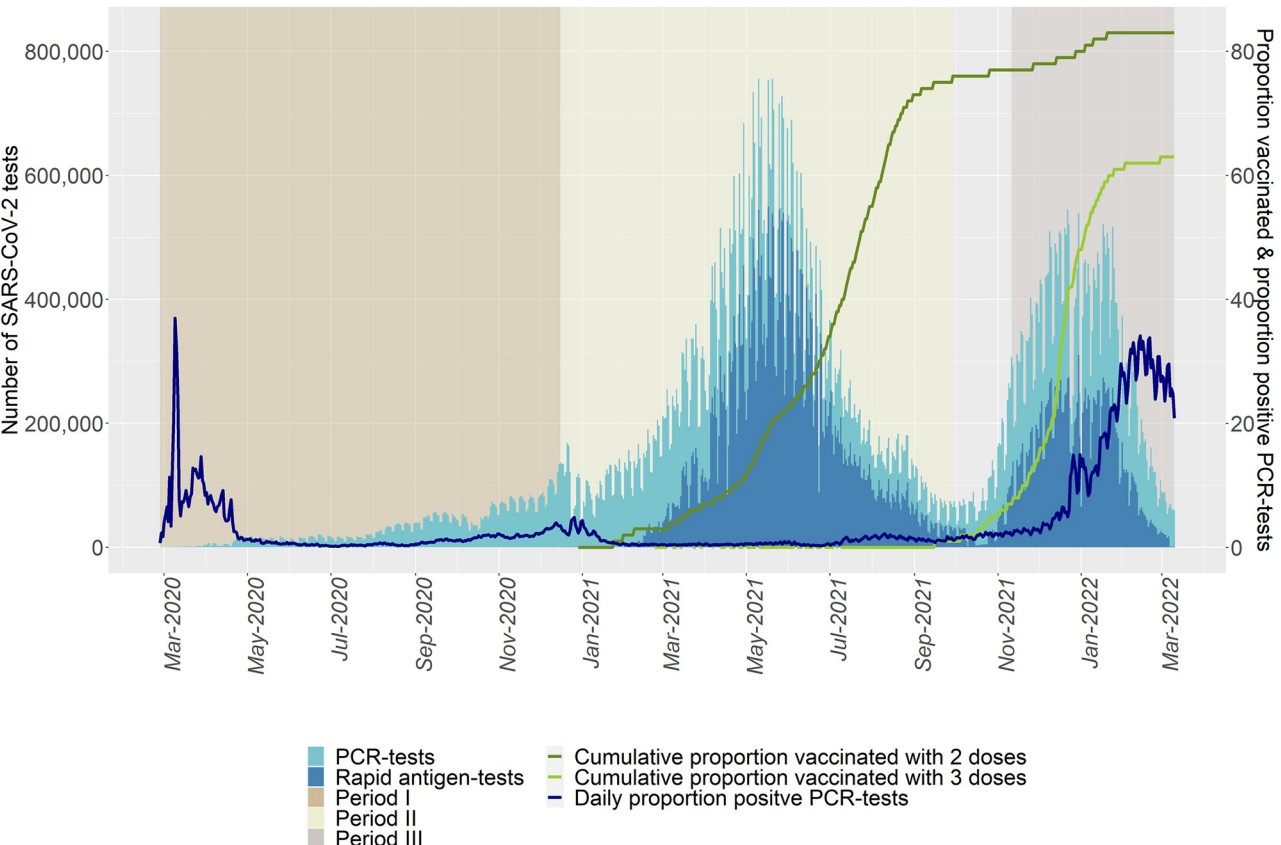

**Fig 1. Number of performed PCR-tests (light blue) and rapid antigen-tests (blue) by sample date, February 27, 2020 to March 10, 2022.**

individuals aged 10–19 years. After SARS-CoV-2 infection, individuals had two times the test rate compared to those without a previous SARS-CoV-2 infection, different from in Periods II and III. Residents of commuter municipalities had the lowest IRR of being PCR-tested (IRR: 0.68) compared to capital municipalities (Fig 3 and S2 Table in S1 File).

In Period II (December 17, 2020 to September 30, 2021), again individuals aged 2–9 years and senior individuals were less likely to be PCR-tested compared to the reference group of individuals aged 10–19 years, while individuals in the age groups from 20–69 had higher IRR's. The highest IRR was observed among individuals aged 40–49 and 50–59 years (IRR: 1.58 and 1.61, respectively). After their second COVID-19 vaccine dose, individuals had 0.46 times the rate of being tested compared to individuals with no or one COVID-19 vaccine dose. After SARS-CoV-2 infection, individuals had 0.72 times the test rate compared to those without a previous SARS-CoV-2 infection. In addition, residents of commuter, metropolitan, provincial and rural municipalities had reduced IRR's, ranging between 0.84 and 0.98 compared to capital municipalities (Fig 3 and S2 Table in S1 File).

In Period III (November 11, 2021 to March 10, 2022), all age groups had lower adjusted IRR's for being PCR-tested compared to individuals aged 10–19 years, except for individuals aged 40–49 years. (IRR: 1.04). Similar to Period II, it was the three older age groups, which were less likely to be PCR-tested. In Period III, individuals vaccinated with two or three COVID-19 vaccine doses were more likely to be PCR-tested compared to individuals who received none or one COVID-19 vaccine dose. For individuals vaccinated with two or three

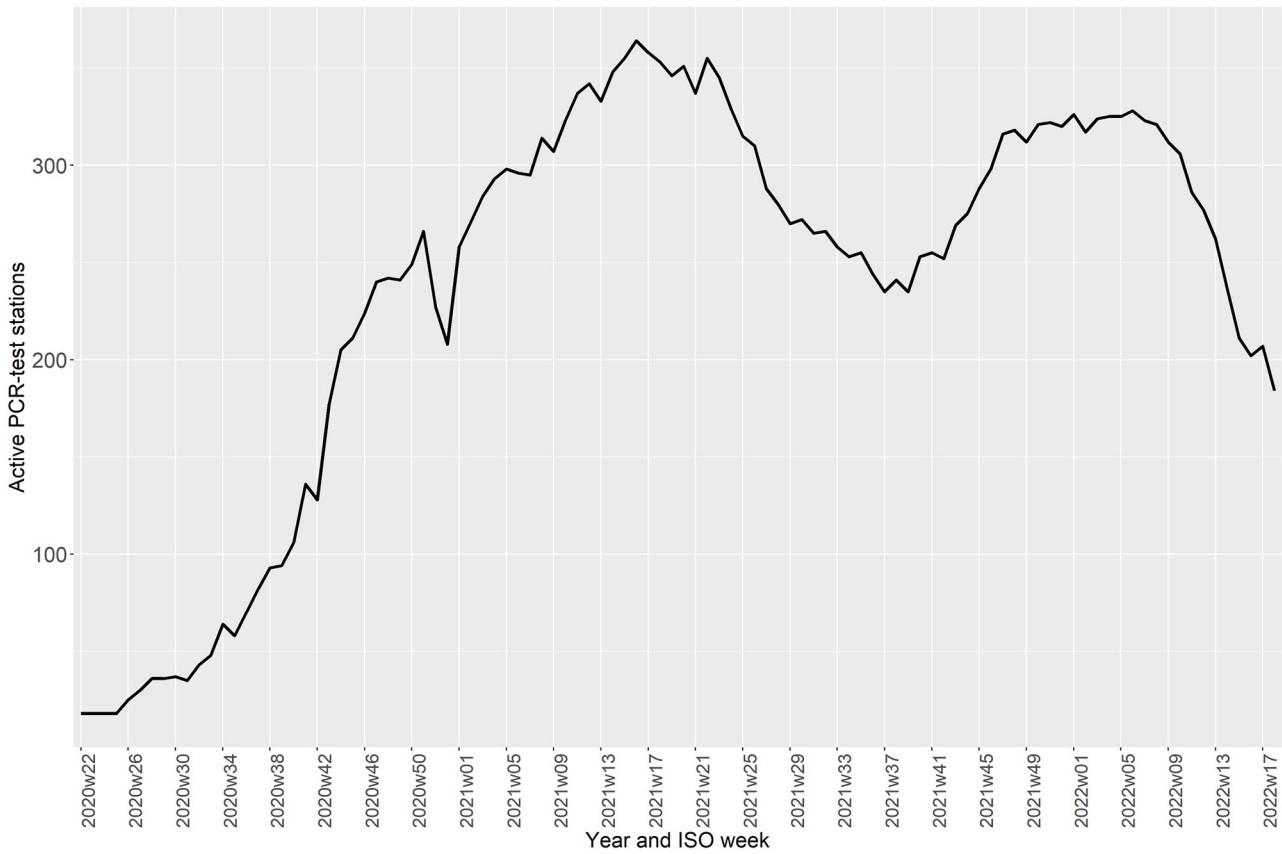

**Fig 2. Active PCR-test stations in Denmark by year and ISO week.** A test station was included as active if $\geq$1 test performed was registered.

doses the IRR was 1.11 and 1.14, respectively. Individuals with a previous SARS-CoV-2 infection were less likely to be PCR-tested (Fig 3 and S2 Table in S1 File).

The results of the sub-analysis including only PCR-tests from the community track, showed the same associations as the main analysis. However, the older age groups were PCR-tested even less in the community track compared to when the community and the healthcare track was analysed as a group. The results are presented in the S3 Table in S1 File.

## Discussion

This study contributes important insight into how the Danish national SARS-CoV-2 test system functioned and how it was received by the population. A total of 63.7 million PCR-tests and 60.0 million rapid antigen-tests were performed between February 27, 2020 and March 10, 2022. We show that a high proportion (90.9%) of the Danish population was PCR-tested at least one time during this period. Although the test system was used by all demographic groups, female sex, younger age, Danish heritage and living in the capital area were all factors positively associated with PCR-testing. The association between COVID-19 vaccination and PCR-testing changed from negative to positive over time. Furthermore, the association between previous infection and PCR-testing changed from positive in Period I to negative in Periods II and III. The lower test frequency among the oldest age groups may possibly be explained by the older population either isolating or finding it difficult to use the digital test booking system or having difficulties transporting themselves to a test station. The high test-

**Table 2. Characteristics of all individuals tested by PCR in Denmark in the three waves.**

| | February 27, 2020 to December 16, 2020 (N-tests 9,007,270) | | December 17, 2020 to September 30, 2021 (N-tests 32,480,632) | | November 11, 2021 to March 10, 2022 (N-tests 20,198,528) | |
|---|---|---|---|---|---|---|
| | Number of tested individuals | Percentage tested | Number of tested individuals | Percentage tested | Number of tested individuals | Percentage tested |
| **Sex** | | | | | | |
| Female | 1.933.842 | 66.8 | 2.281.866 | 78.5 | 2.356.071 | 80.9 |
| Male | 1.735.310 | 60.5 | 2.138.126 | 74.2 | 2.219.872 | 77.0 |
| **Age groups (years)** | | | | | | |
| 2–9 | 236.859 | 54.5 | 306.105 | 70.4 | 371.970 | 85.2 |
| 10–19 | 453.924 | 75.1 | 513.454 | 85.2 | 549.698 | 92.2 |
| 20–29 | 504.126 | 73.0 | 580.245 | 84.2 | 564.113 | 83.2 |
| 30–39 | 472.519 | 70.4 | 573.056 | 83.8 | 584.858 | 83.9 |
| 40–49 | 430.383 | 69.8 | 512.713 | 84.1 | 525.631 | 86.4 |
| 50–59 | 474.821 | 67.1 | 574.788 | 82.2 | 583.567 | 84.5 |
| 60–69 | 432.387 | 62.4 | 550.825 | 78.2 | 561.048 | 78.8 |
| 70–79 | 309.294 | 53.4 | 392.578 | 67.7 | 398.063 | 68.6 |
| 80–89 | 236.000 | 47.5 | 287.996 | 57.0 | 294.683 | 57.4 |
| ≥90 | 118.839 | 44.2 | 128.232 | 46.2 | 142.312 | 49.9 |
| **Vaccinated with ≥2 doses before March 10 2022** | | | | | | |
| Yes | 3.163.367 | 65.9 | 3.782.195 | 78.3 | 3.829.514 | 79.8 |
| No | 505.785 | 52.6 | 637.797 | 66.5 | 746.429 | 74.6 |
| **Heritage** | | | | | | |
| Danish | 3.108.265 | 64.5 | 3.715.129 | 76.9 | 3.871.688 | 80.2 |
| Non-west | 334.796 | 63.0 | 419.380 | 76.9 | 416.391 | 74.6 |
| West | 225.352 | 58.8 | 284.594 | 69.9 | 286.952 | 70.0 |
| Unknown | 739 | 2.3 | 889 | 65.4 | 912 | 67.6 |
| **Type of area (municipality)** | | | | | | |
| Capital Municipalities | 1.102.768 | 69.6 | 1.283.236 | 80.4 | 1.284.965 | 80.2 |
| Commuter Municipalities | 537.020 | 57.5 | 689.703 | 73.7 | 728.313 | 77.7 |
| Metropolitan Municipalities | 505.063 | 65.9 | 608.358 | 79.0 | 614.600 | 79.6 |
| Provincial Municipalities | 802.215 | 60.6 | 1.006.438 | 75.8 | 1.056.173 | 79.4 |
| Rural Municipalities | 722.086 | 62.4 | 832.257 | 71.9 | 891.892 | 77.2 |
| Unknown | 1.933.842 | 66.8 | 2.281.866 | 78.5 | 2.356.071 | 80.9 |

frequency among children is likely associated with this group being more often unvaccinated. Vaccination was not offered before the age of 6 years and overall vaccination coverage among individuals younger than 16 years was 37%, less than for adults (86%), by end of study. The fact that persons with recent immigration background had lower test rates is interesting; it might in part be due to language barriers or perhaps with a tendency to be disenfranchised and less likely to follow government guidelines.

In the study period between December 17, 2020 and September 30, 2021, people vaccinated with two COVID-19 vaccine doses were less likely to be PCR-tested compared to those given none or just a single COVID-19 vaccine dose (IRR: 0.44). However, in the study period between November 11, 2021 and March 10, 2022, those vaccinated with two or three COVID-19 vaccine doses were more often PCR-tested despite public health authorities having encouraged more frequent testing for the unvaccinated. Compared to those given none or just a single COVID-19 vaccine dose, the IRR was 1.11 and 1.14 for two and three vaccine doses,

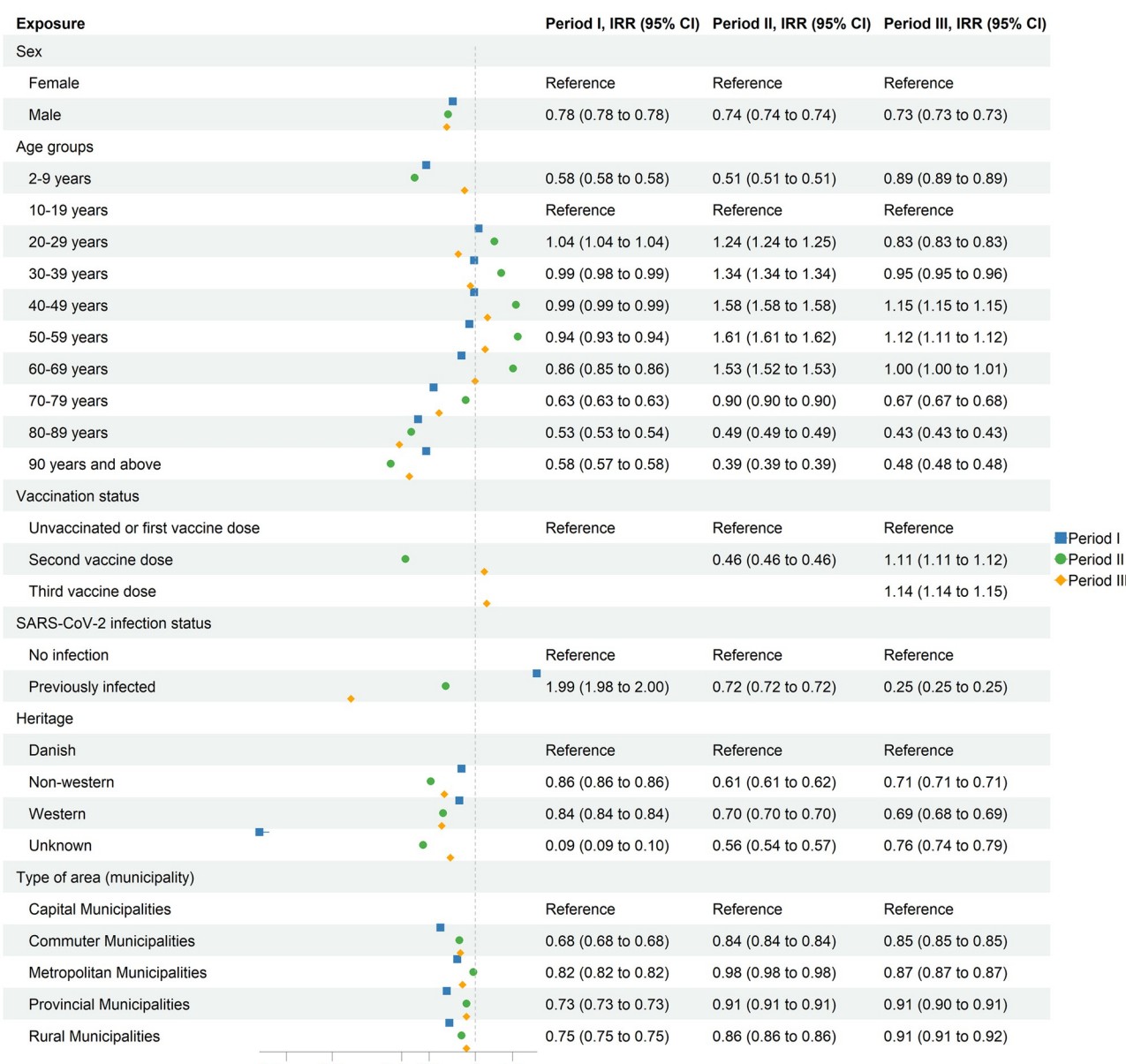

**Fig 3. Adjusted IRR for PCR-testing by sex, age, vaccination status, infection status, heritage and type of area in three study periods.** IRR were adjusted for age group, sex, vaccination status, SARS-CoV-2 infection status, heritage, and type of area.

respectively. The Omicron SARS-CoV-2 variant was circulating in this period and it was known that vaccine effectiveness was lower against this variant [10]. This might explain why individuals were tested regardless of their vaccination status. Furthermore, the unvaccinated group changes over time and between the two periods. Unvaccinated people may have been engaging less with the health care system overall, especially in Period III where all individuals were offered vaccination.

Denmark performed more tests per inhabitants than other comparable countries. To illustrate, between February 27, 2020 and March 10, 2022, the mean number of PCR-tests per day per thousand persons was 14.4 in Denmark, 3.0 in Sweden, 2.8 in Norway, 9.6 in the UK and

2.4 in the Netherlands [3]. The Danish national SARS-CoV-2 test system represented a huge infrastructure investment, but also proved very valuable for surveillance and control of the epidemic. Testing as many individuals as possible, tracking infected individuals, and tracing their contacts was deemed in Denmark as an effective strategy to reduce the spread of the virus. Timely and detailed surveillance data provided a foundation for risk assessments that informed public health regulations and thus decisions on restrictions such as mask mandates and launch or release of lockdowns could be made on a foundation of scientific facts. Daily geographic detailed summary results from the tests was made publicly available via web-based dashboards that were shared with stakeholders and widely reported in the press. This rapid and open publication of surveillance data was likely also an important factor in maintaining the population's confidence in the public health authorities.

The World Health Organization (WHO) stated that testing was a critical element to the overall prevention and control strategy for COVID-19 [11]. The WHO recommends that all individuals meeting the case definition for COVID-19, irrespective of vaccination or disease history, be tested for SARS-CoV-2 [11]. However, testing of asymptomatic individuals can be informative in instances such as follow up of contacts of confirmed or probable cases or testing of health care and long-term care facility workers that are frequently exposed [11]. Notwithstanding, widespread testing of asymptomatic populations, including self-testing, is not recommended by the WHO, based on lack of evidence on impact and cost-effectiveness of such approaches and the concern that this approach risks diverting resources from higher priority testing indications [11]. Thus, the Danish test system thus goes beyond what WHO recommended. The available evidence of the effect of mass testing is sparse. A Swiss single-centre cohort study compared two testing strategies (restricted and extended strategy) [12]. They concluded that widespread testing is crucial to understand and control the spread of infection, and to maximize identification of infected people. Access to free testing was found to be essential, not only to achieve infection control, but also to eliminate any discrimination between the different layers of society [12].

The Danish test system relied on several features/strengths that enabled the large-scale testing for SARS-CoV-2. The system leveraged on an existing infrastructure, including information and communication technologies, which facilitated the centralized and uniform lab set-up and the ability for users to safely log in and book test time slots and see the test results via computer/smartphone as well as the ability to individually link data on all residents in Denmark across the nationwide high-quality registries. The low turnaround times were also a strength as the rapid response made it possible for users to make informed decisions based on their test results, including mitigation of further spread of SARS-CoV-2. Another strength was that it was mandatory for providers performing SARS-CoV-2 testing to report electronically to MiBa [5]. A limitation was that rapid antigen self-tests for SARS-CoV-2 infection were not included in the national surveillance. However, self-tests were introduced into Denmark late and played a smaller role than in many other countries due to the wide availability of PCR-tests and those testing positive by self-tests were advised to take a follow-up test in the national PCR-test system. Although we adjusted the Poisson regression models for potential confounders (sex, age, vaccination status, infection status, heritage and type of area), we cannot rule out that unmeasured confounders such as time varying COVID-19 guidelines, distance to test station, socioeconomic factors, and health behaviour may have affected the results. Further research should include qualitative research to examine barriers for PCR-testing. Furthermore, an assessment of the impact and the cost effectiveness of the system is beyond the scope of the present paper.

The ability to effectively track and trace variants of concern was considered valuable. For example, it was considered critical for an early and stepwise reopening of the Danish society in

April 2020 and it also made it possible to delay community spread of the Delta variant by 10 weeks. This was of major importance as it provided sufficient time to vaccinate vulnerable groups thus mitigating the impact of the Delta variant.

Another benefit of the Danish testing infrastructure was its use for scientific studies. Because the test information was person-identifiable and linkage with national health and administrative registers were possible, the test-system also became a resource for epidemiological studies. The different use of the test system by vaccine status has implications for field studies of vaccine effectiveness. For example, a higher testing activity among vaccine recipients may lead to underestimating vaccine effectiveness since case ascertainment will increase in this group. Likewise, in the estimation of immunity following natural infections, different test behavior after a positive PCR will influence the outcome. The Danish test stations were also utilised to also conduct national sero-epidemiological study series [13]. Moreover, the person-identifiable, centrally registered test data in combination with Danish health and administrative registers offering data-linkage, proved an important epidemiological data resource and was used for a large series of studies providing information, often of international value, during the epidemic. Subsequently, studies on population protection [14, 15], vaccine effectiveness [10, 16, 17], virulence of variants [18, 19], epidemical dynamics [20], long COVID [21] and forecasting and risks associated with vulnerable patient or community groups [22–26] were produced using the data from the study period. With the current paper, we hope to provide a further basis for such studies to come.

It should be recognised that testing patterns depend on factors besides the actual service provided. Throughout the pandemic various restrictions were implemented and lifted depending on the development of the pandemic and the developing understanding of the transmission of SARS-CoV-2 [27]. During certain periods a negative PCR-test was necessary in order take part in civil life (e.g. access to sport/public institutions, restaurants, public cultural activities or foreign travel) [27]. Further, staff at hospitals, nursing homes and home care have actively been encouraged to be voluntarily tested at their workplace several times a week for prolonged periods. Similarly, school children and students were tested weekly or biweekly in some periods; when antigen-testing was used those testing positive were advised to be re-tested again by PCR-test [27]. These recommendations and requirements definitely played a role in testing patterns but is beyond the scope of this study.

In conclusion, the results demonstrate that the extensive, free-of-charge, national SARS-CoV-2 test system in Denmark was well received by the majority of Danish population. However, demographic differences in the test uptake existed suggesting challenges to health equity in the society and possible biases in studies related to SARS-CoV-2. In the future, it might be relevant to focus on increasing the test uptake in individuals aged 70 years or above and unvaccinated individuals.

## Supporting information

**S1 File.**
(DOCX)

## Acknowledgments

We would like to thank the staff in all the test centers, at the ten Departments of Clinical Microbiology in Denmark, and at the many departments of the SSI taking part in setting up and running the test system (including the Virology Laboratory, TestCenter Denmark, Campus Services, the National Biobank, Infections Disease Epidemiology and Prevention, and the

Data Integration and Analysis Secretariat). The authors thank Dr. Martin Collin Fjordholt for critical reading of the manuscript.

## Author Contributions

**Conceptualization:** Mie Agermose Gram, Nina Steenhard, Arieh Sierra Cohen, Steen Ethelberg.

**Data curation:** Mie Agermose Gram.

**Formal analysis:** Mie Agermose Gram.

**Methodology:** Mie Agermose Gram, Steen Ethelberg.

**Project administration:** Steen Ethelberg.

**Resources:** Mie Agermose Gram, Nina Steenhard, Arieh Sierra Cohen, Anne-Marie Vangsted, Kåre Mølbak, Thøger Gorm Jensen, Christian Holm Hansen, Steen Ethelberg.

**Writing – original draft:** Mie Agermose Gram, Steen Ethelberg.

**Writing – review & editing:** Mie Agermose Gram, Nina Steenhard, Arieh Sierra Cohen, Anne-Marie Vangsted, Kåre Mølbak, Thøger Gorm Jensen, Christian Holm Hansen, Steen Ethelberg.

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
