## [Decision Letter · Decision Letter 0]

3 Mar 2023

PONE-D-23-03172Patterns of testing in the extensive Danish national SARS-CoV-2 test set-upPLOS ONE

Dear Dr. Gram, Thank you for submitting your manuscript to PLOS ONE. After careful consideration, we feel that it has merit but does not fully meet PLOS ONE’s publication criteria as it currently stands. Therefore, we invite you to submit a revised version of the manuscript that addresses the points raised during the review process. The manuscript will be re-evaluated after minor revisions as suggested by the reviewers.

We look forward to receiving your revised manuscript.

Kind regards,

Om Prakash Choudhary, Ph.D.

Academic Editor

PLOS ONE

Journal Requirements:

"..Many structures in Danish society contributed to establishing and running the test centres, special contributions were given by the Novo Nordic Foundation (Grant number: NNF20SA0063854), Novo Nordisk Denmark (providing staff), McKinsey (sourcing) and the Danish Military (logistical support)."

"The authors received no specific funding for this work."

5. Please upload a new copy of Figures 1 and 2 as the detail is not clear. Please follow the link for more information:

https://blogs.plos.org/plos/2019/06/looking-good-tips-for-creating-your-plos-figures-graphics/

https://blogs.plos.org/plos/2019/06/looking-good-tips-for-creating-your-plos-figures-graphics/

Reviewers' comments:

Reviewer's Responses to Questions

**Comments to the Author**

1. Is the manuscript technically sound, and do the data support the conclusions?

Reviewer #1: Yes

Reviewer #2: Yes

2. Has the statistical analysis been performed appropriately and rigorously? 

Reviewer #1: Yes

Reviewer #2: I Don't Know

3. Have the authors made all data underlying the findings in their manuscript fully available?

Reviewer #1: Yes

Reviewer #2: Yes

4. Is the manuscript presented in an intelligible fashion and written in standard English?

Reviewer #1: Yes

Reviewer #2: Yes

5. Review Comments to the Author

Reviewer #1: The manuscript titled "Patterns of testing in the extensive Danish national SARS-CoV-2 test set-up" is an interesting study.

I appreciate the authors' topic. The authors clearly defined the research questions in the background section. However, the following comments need to be addressed, and the authors should improve their manuscript in the following aspects:

1. Please improve the quality of the manuscript.

2. This research is important and useful because it will help improve the COVID-19 national health system in every country. This will make it easier to find out things like who is infected or has no symptoms.

Has a comparable study been done in other countries? I think it will help improve the adoption of health programs in the future.

And on the basis of that, a unique regulation was established. Mention it if the answer is positive.

I am aware that the number of samples required for statistical analysis is enormous, but you could summarize the key points of the data in Table 3 in a graph in order to prevent generalization and scattering and to facilitate content transfer for better understanding.

Reviewer #2: Dear authors, overall, the manuscript is well written & presents an excellent results, I don't have any scientific revision, only I have comments about figures quality it should be replaced with a good ones.

Best wishes

6. PLOS authors have the option to publish the peer review history of their article (what does this mean?). If published, this will include your full peer review and any attached files.

Reviewer #1: No

Reviewer #2: No

---

## [Author Response · Author response to Decision Letter 0]

11 Apr 2023

Response to editor and reviewers

Thank you for taking the time to comment on our manuscript. Your comments have been very helpful to improve the manuscript. The information has been added to the manuscript and a point-by-point response is included below. 

Journal Requirements

Response: Thank you for sharing the templates. The manuscript has been revised to meet the style requirements. 

"..Many structures in Danish society contributed to establishing and running the test centres, special contributions were given by the Novo Nordic Foundation (Grant number: NNF20SA0063854), Novo Nordisk Denmark (providing staff), McKinsey (sourcing) and the Danish Military (logistical support)."

"The authors received no specific funding for this work."

Response: Thank you for pointing this out. We have removed the funding-related information from the manuscript. The authors received no funding for the work presented in this paper. However, many structures in Danish society contributed to establishing and running the test centres, special contributions were given by the Novo Nordic Foundation (Grant number: NNF20SA0063854), Novo Nordisk Denmark (providing staff), McKinsey (sourcing) and the Danish Military (logistical support). Can we include this information in the supporting file? 

Response: Thank you. The underlying dataset, which consists of more than 120 million data rows, contains person identifiable information concerning most of the Danish population. We are not in a position to share this and we believe that we have provided an explanation for this already. Should we have misunderstood, we’re of course happy to discuss what to do. The text we previously provided reads: 

“The data material used involve information on every person living in Denmark within the study period. Data cannot be shared publicly because they may only be accessed and register-coupled within a secure data analysis environment. However, de-identified participant-level data are available for Institutional Data access to members of the scientific and medical community for non-commercial use only. Applications should be submitted to Forskerservice (https://sundhedsdatastyrelsen.dk/da/forskerservice) at The Danish Health Data Authority, where they will be reviewed on the basis of relevance and scientific merit. Data are available now, with no defined end date.”

Response: The ethic statement has been moved to the Methods section and deleted from any other section. 

5. Please upload a new copy of Figures 1 and 2 as the detail is not clear. Please follow the link for more information:

https://blogs.plos.org/plos/2019/06/looking-good-tips-for-creating-your-plos-figures-graphics/

https://blogs.plos.org/plos/2019/06/looking-good-tips-for-creating-your-plos-figures-graphics/

Response: Thank you for giving us the opportunity to improve the quality of the figures. The figures have been re-created in R studio instead of excel, which makes it easier to adjust the dimensions and resolution. Please let us know if the new version of the figures appears blurry. 

Response: We have ensured that the reference list is complete and correct. 

Review Comments to the Author

Reviewer #1: The manuscript titled "Patterns of testing in the extensive Danish national SARS-CoV-2 test set-up" is an interesting study.

I appreciate the authors' topic. The authors clearly defined the research questions in the background section. However, the following comments need to be addressed, and the authors should improve their manuscript in the following aspects:

1. Please improve the quality of the manuscript.

Response: We believe that, with the number of changes we’ve made to the manuscript, including the new figure, the overall quality of the manuscript has improved considerably. We hope by this to have responded adequately to this comment by the reviewer. 

2. This research is important and useful because it will help improve the COVID-19 national health system in every country. This will make it easier to find out things like who is infected or has no symptoms.

Has a comparable study been done in other countries? I think it will help improve the adoption of health programs in the future.

And on the basis of that, a unique regulation was established. Mention it if the answer is positive.

I am aware that the number of samples required for statistical analysis is enormous, but you could summarize the key points of the data in Table 3 in a graph in order to prevent generalization and scattering and to facilitate content transfer for better understanding.

Response: To our knowledge no comparable studies have been done in other countries. We appreciate your suggestion of visualizing the data in Table 3 which we have followed. The results in Table 3 are now shown in Fig 3. We believe this new figure provide a better overview of the results of the analysis. Table 3 has been moved to Supplementary and appears as S3 Table. We decided to keep the table in supplementary because it adds additional information of number of events and person-years. 

Reviewer #2: Dear authors, overall, the manuscript is well written & presents an excellent results, I don't have any scientific revision, only I have comments about figures quality it should be replaced with a good ones. Best wishes.

Response: Thank you so much for your comments. The figures have been re-created in R studio instead of excel, which makes it easier to adjust the dimensions and resolution to ensure high quality.

---

## [Decision Letter · Decision Letter 1]

3 Jul 2023

PONE-D-23-03172R1Patterns of testing in the extensive Danish national SARS-CoV-2 test set-upPLOS ONE

Dear Dr. Gram,

Thank you for submitting your manuscript to PLOS ONE. After careful consideration, we feel that it has merit but does not fully meet PLOS ONE’s publication criteria as it currently stands. Therefore, we invite you to submit a revised version of the manuscript that addresses the points raised during the review process.

We look forward to receiving your revised manuscript.

Kind regards,

Marwan Osman

Academic Editor

PLOS ONE

Journal Requirements:

Reviewers' comments:

Reviewer's Responses to Questions

**Comments to the Author**

1. If the authors have adequately addressed your comments raised in a previous round of review and you feel that this manuscript is now acceptable for publication, you may indicate that here to bypass the “Comments to the Author” section, enter your conflict of interest statement in the “Confidential to Editor” section, and submit your "Accept" recommendation.

Reviewer #3: All comments have been addressed

2. Is the manuscript technically sound, and do the data support the conclusions?

Reviewer #3: Yes

3. Has the statistical analysis been performed appropriately and rigorously? 

Reviewer #3: Yes

4. Have the authors made all data underlying the findings in their manuscript fully available?

Reviewer #3: Yes

5. Is the manuscript presented in an intelligible fashion and written in standard English?

Reviewer #3: Yes

6. Review Comments to the Author

Reviewer #3: The authors provide interesting insights on the use of COVID-19 test during the pandemic. The topic is overall interesting, but some clarification need to be addressed:

1) Could you specify if there were mandatory testing before attending mass gathering events, especially during the first wave

2) Authors have identified two tracks for testing : healthcare track/community track. It would be interesting to report results stratified by tracks, to see the weight of each track in the overall testing policy

3) Is there any ways to combine the database of PCR and Rapid antigen testing? I would interesting to have all analyses pooled, esp:

- How many people did not have neither a PCR nor a Rapid antigen test

- What is the proportion, at individual level, of test being done by PCR and Rapid antigen, if those who had rapid antigen also had PCR to confirm the diagnosis. In the latter case, if there were discrepancies.

- Authors barely mentionned the rate of positive PCR tests (only in a figure), it would be of interest to have a more detailed look

4) Discussion is too long, with comments that are not related to results being presented

5) The association between vaccination and testing should be more elaborated : those having > 2 doses may be person at risks (immmunocompromised), and thus be more tested in the context of their healthcare plan (hence the value of looking at the different testing tracks).

7. PLOS authors have the option to publish the peer review history of their article (what does this mean?). If published, this will include your full peer review and any attached files.

Reviewer #3: No

---

## [Author Response · Author response to Decision Letter 1]

10 Jul 2023

Journal Requirements

Response: We have reviewed the reference list to ensure that it is complete and correct.

Review Comments to the Author

1) Could you specify if there were mandatory testing before attending mass gathering events, especially during the first wave

Response: Thank you for your valuable comment. We appreciate your consideration of our findings. We agree that test requirements played an important role in the testing patterns. Throughout the pandemic various restrictions including test requirements were implemented and lifted depending on the development of the pandemic. Our colleagues have done a great work describing the restrictions in Figure 1 in a published study (Munch PK, Espenhain L, Hansen CH, Krause TG, Ethelberg S. Case-control study of activities associated with SARS-CoV-2 infection in an adult unvaccinated population and overview of societal COVID-19 epidemic counter measures in Denmark. PLoS One. 2022;17(11):e0268849. doi: 10.1371/journal.pone.0268849). In our study we briefly mention the restrictions and cite our colleagues study for further information. “During certain periods a negative PCR-test was necessary in order take part in civil life (e.g. access to sport/public institutions, restaurants, public cultural activities or foreign travel) [27]. Further, staff at hospitals, nursing homes and home care have actively been encouraged to be voluntarily tested at their workplace several times a week for prolonged periods. Similarly, school children and students were tested weekly or biweekly in some periods [27].”

2) Authors have identified two tracks for testing: healthcare track/community track. It would be interesting to report results stratified by tracks, to see the weight of each track in the overall testing policy

Response: Thank you for your suggestion. As described in the manuscript, the proportion of PCR-test performed in the healthcare track was only 16.6%. In the supplementary material, we included the results of a sub-analysis including only PCR-tests from the community track, which showed the same associations as the main analysis. However, the older age groups were PCR-tested less in the community track compared to when the community and the healthcare track was analysed as a group (S3 Table). 

3) Is there any ways to combine the database of PCR and Rapid antigen testing? I would interesting to have all analyses pooled, esp:

- How many people did not have neither a PCR nor a Rapid antigen test

- What is the proportion, at individual level, of test being done by PCR and Rapid antigen, if those who had rapid antigen also had PCR to confirm the diagnosis. In the latter case, if there were discrepancies.

- Authors barely mentioned the rate of positive PCR tests (only in a figure), it would be of interest to have a more detailed look

Response: Thank you for your valuable comment. During the epidemic, the PCR-test results were the primary basis for surveillance and national incidence figures. Moreover, PCR-tests have been primarily utilized in epidemiological studies. Individuals who tested positive by rapid antigen test were advised to have the infection confirmed in a subsequent PCR test. Therefore, we decided not to include rapid antigen-tests in all analyses. We added information about the proportion of individuals never PCR- or rapid antigen-tested (5.4%) to the manuscript (page 10). Lastly, we are not quite certain about what extra details of the positive PCR-tests the reviewer find relevant to include in the manuscript. 

4) Discussion is too long, with comments that are not related to results being presented

Response: We thank the reviewer for this comment. However, we are not quite certain where precisely the reviewer suggests to shorten the Discussion. We feel, naturally, that the Discussion is of adequate length and that it covers the presented results. Also, this point had not previously been raised. Since decisions concerning the length of the manuscript also involve journal policy, we might suggest to keep the discussion at its present length and leave it to the editorial office, to decide if it should be rewritten.

5) The association between vaccination and testing should be more elaborated: those having > 2 doses may be person at risks (immmunocompromised), and thus be more tested in the context of their healthcare plan (hence the value of looking at the different testing tracks).

Response: We appreciate your comments and the time you have dedicated to review our work. In response to the specific point raised, we would like to address that individuals aged 18 years or above were offered 3 vaccine doses and the vaccination coverage in the total study population was above 60% (figure 1). Therefore, we do not believe that individuals having more than two doses are at higher risk or immunocompromised. We hope that you agree.

---

## [Decision Letter · Decision Letter 2]

13 Jul 2023

Patterns of testing in the extensive Danish national SARS-CoV-2 test set-up

PONE-D-23-03172R2

Dear Dr. Gram,

We’re pleased to inform you that your manuscript has been judged scientifically suitable for publication and will be formally accepted for publication once it meets all outstanding technical requirements.

Kind regards,

Marwan Osman

Academic Editor

PLOS ONE

---

## [Editor Report · Acceptance letter]

17 Jul 2023

PONE-D-23-03172R2 

Patterns of testing in the extensive Danish national SARS-CoV-2 test set-up 

Dear Dr. Gram:

I'm pleased to inform you that your manuscript has been deemed suitable for publication in PLOS ONE. Congratulations! Your manuscript is now with our production department. 

Kind regards, 

on behalf of

Dr. Marwan Osman 

Academic Editor

PLOS ONE